# The Synthetic Collagen-Binding Peptide NIPEP-OSS Delays Mouse Myeloma Progression

**DOI:** 10.3390/cancers15092473

**Published:** 2023-04-26

**Authors:** Syed Hassan Mehdi, Austin C. Gentry, Jue-Yeon Lee, Chong-Pyoung Chung, Donghoon Yoon

**Affiliations:** 1Myeloma Center, The University of Arkansas for Medical Sciences, Little Rock, AR 72205, USA; 2Research Institute, NIBEC Co., Ltd., 174 Yulgok-ro, Jongno-gu, Seoul 03170, Republic of Korea

**Keywords:** multiple myeloma, NIPEP-OSS, bone anabolic agent, bone mineralization, multiple myeloma bone disease

## Abstract

**Simple Summary:**

Multiple myeloma is a plasma cell cancer with bone destruction and is still considered an incurable disease despite advancements in its treatment. Over 90% of patients experience bone destruction during the disease’s course. NIPEP-OSS has a targeted osteogenic activity with a broad safety margin. Therefore, we repurposed NIPEP-OSS for multiple myeloma bone disease (MMBD). The present study demonstrated that NIPEP-OSS delays mouse myeloma progression via bone formation in MMBD mouse models.

**Abstract:**

Multiple myeloma (MM) is the second most common hematological malignancy. It is a clonal B-cell disorder characterized by the proliferation of malignant plasma cells in the bone marrow, the presence of monoclonal serum immunoglobulin, and osteolytic lesions. An increasing amount of evidence shows that the interactions of MM cells and the bone microenvironment play a significant role, suggesting that these interactions may be good targets for therapy. The osteopontin-derived collagen-binding motif-bearing peptide NIPEP-OSS stimulates biomineralization and enhances bone remodeling dynamics. Due to its unique targeted osteogenic activity with a broad safety margin, we evaluated the potential of NIPEP-OSS for anti-myeloma activity using MM bone disease (MMBD) animal models. In a 5TGM1-engrafted NSG model, the survival rates of the control and treated groups were significantly different (*p* = 0.0014), with median survival times of 45 and 57 days, respectively. The bioluminescence analyses showed that myeloma slowly developed in the treated mice compared to the control mice in both models. NIPEP-OSS enhanced bone formation by increasing biomineralization in the bone. We also tested NIPEP-OSS in a well-established 5TGM1-engrafted C57BL/KaLwRij model. Similar to the previous model, the median survival times of the control and treated groups were significantly different (*p* = 0.0057), with 46 and 63 days, respectively. In comparison with the control, an increase in p1NP was found in the treated mice. We concluded that NIPEP-OSS delays mouse myeloma progression via bone formation in MMBD mouse models.

## 1. Introduction

Mineralization, the deposition process of calcium, phosphate, and other ions in the bone matrix, is an essential process for generating bone and is orchestrated by the regulation of protein-based extracellular matrix components, such as non-collagenous proteins and collagen fibrils. The development of biomaterials inducing active biomineralization for bone regeneration is an active area of research with high demand in clinical regenerative medicine. A complex structure of bone requires tailored bioactive scaffolds and highly precise biomineralization. Several biocompatible materials, such as collagen and synthetic polymers, have been employed as scaffolds [1,2,3,4]. The surface modification of these materials with synthetic adhesive ligands improved the efficacy of bone regeneration [5]. Collagen is a major extracellular matrix protein. Collagen fibrils in bone have a high binding affinity to bone sialoprotein, dentin matrix protein 1, phosphophory, and osteopontin (OPN), which stimulates mineral deposition [6,7,8,9]. In addition, collagen may induce apatite formation [10,11,12]. Thus, combining collagen with active protein and matrix protein can induce mineral creation, causing bone formation in vivo. However, clinical applications have been hindered due to its short half-life, immune system side effects, and high cost. Instead, an oligopeptide that stably binds to collagen was designed to provide collagen scaffolding activity from the OPN binding motif [5]. The OPN is known to bind collagen, induce mineral creation, and regulate bone formation [13,14,15]. An OPN-derived collagen-binding motif (CBM)-bearing peptide that has specific non-covalent interactions between scaffolding collagen polymers and bioactive oligopeptides was generated, actively stimulating biomineralization and enhancing bone remodeling dynamics [5]. Furthermore, the CBM peptide selectively induced osteogenic differentiation, while reducing adipogenic differentiation in human mesenchymal stem cells and inducing bone formation in ovariectomized mice [16]. The osteopontin-derived CBM-bearing peptide has been commercialized as NIPEP-OSS by NIBEC Co. (Seoul, Republic of Korea). It has an osteogenic agent with a broad safety margin which can be rapidly distributed to skeletal tissue, increasing bone formation [16].

Bone remodeling is firmly controlled through an extensive variety of signaling pathways [17,18] which couple bone resorption by osteoclasts [19,20] and bone formation by osteoblasts [21,22,23,24,25,26]; imbalances of these mechanisms result in bone diseases [27]. Multiple myeloma (MM) is the second most common blood cancer derived from the extraproliferation of malignant plasma cells in the bone marrow. In 84% of patients, skeletal lesions develop during the course of the disease [28]. MM is highly associated with bone destruction. Osteoclasts have been shown to enhance bone destruction and support myeloma progression, while osteoblasts have been shown to enhance bone formation and inhibit myeloma growth [29]. In MM, the activities of osteoclasts are up-regulated while the osteoblast activity is suppressed. Several studies have suggested the induction of osteoblast differentiation/activation as one of the essential first steps of biomineralization [30,31,32]. In addition, our retrospective patient evaluation study found extensive bone remineralization of large pelvic lytic lesions after MM therapy [33]. Therefore, we have sought bone anabolic agents (BAA) that induce bone formation and ultimately suppress MM progression with minor toxicity. We recently improved an MM animal model [34]. In this model, we were able to detect the early onset of MM and fairly uniform tumor progression. Severe bone disease was developed in this model compared to the conventional C57BL/KaLwRij model, making this model a better tool for screening therapeutic candidates.

In this study, we test whether the novel BAA NIPEP-OSS inhibits myeloma bone disease progression using a newly developed MMBD model and a conventional model. We also evaluate if such a therapeutic effect was mediated by bone formation.

## 2. Materials and Methods

### 2.1. Cell Culture and Maintenance

The 5TGM1-Luc cells were kindly shared by Dr. Oyajobi at the University of Texas Health Science Center (San Antonio, TX, USA). They were cultured at 37 °C at 5% CO_2_ in RPMI 1640 medium supplemented with 15% fetal bovine serum (FBS) + 1× penicillin/streptomycin + 1× Glutmax (Gibco, Life Technologies, Carlsbad, CA, USA). The 5TGM1-Luc cells were washed and counted via Cellometer Mini (Nexcelom, Lawrence, MA, USA) using the trypan blue exclusion method before inoculating them into the mice [34]. The targeted viability was greater than 90%.

### 2.2. Animal Experimentation

NSG and C57BL/KaLwRij strain mice were housed and bred at the University of Arkansas for Medical Sciences (UAMS) Animal Facility. All animal procedures stated in the current study were approved by the UAMS IACUC. One million cells (>90% viability) in 100 µL PBS were injected into an equal number of genders of 8–12-week-old mice via tail vein. To ensure myeloma cell engraftment, we performed bioluminescence imaging weekly. The mice that showed no signal until three weeks post-injection were removed from the study.

### 2.3. Bioluminescence Imaging

All the mice were weekly imaged from 1st~8/9th week of post-inoculation to access the tumor burden. For this, mice were anesthetized using 3% isoflurane and given D-luciferin (1.5 mg/mouse, Perkin Elmer, Waltham, MA, USA) via intraperitoneal injection. Mice were imaged in an IVIS Imaging System 200 Series (Perkin Elmer, Waltham, MA, USA). Images were obtained by auto exposure, and to acquire the total flux (*p*/*s*), a region of interest was created to cover the whole body using Living Image 4.7.4 software.

### 2.4. p1NP Level Measurement

By retro-orbital bleeding, 50 µL of blood was collected and separated by centrifugation at 8000 rpm for 5 min. The plasma was transferred into other vials and stored at −80 °C for the assay. Serum p1NP levels were analyzed using a mouse p1NP ELISA kit (R&D Systems, Minneapolis, MN, USA) according to the manufacturer’s instructions.

### 2.5. Bone Histomorphometric and Dexa Analysis for Spine

After meeting the endpoint criteria, mice were sacrificed and stored at −20 °C (>2 days). After thawing them at room temperature, the skin was removed, then scanned by a PIXImus bone densitometer with on-board PIXImus software (G.E. Lunar, Madison, WI, USA). Bone mineral content and bone mineral density of the vertebrae were evaluated by drawing a region of interest over the vertebrae and left/right femur. Then, the spine was extracted. The Lumbar vertebrae (L1–L6) were scanned using a microCT40 (SCANCO Medical, Bassersdorf, Switzerland). The trabecular regions of the vertebral body were selected for histomorphometric bone analyses. The analyses provided the main histomorphometry parameters: bone volume/total volume (BV/TV, %), trabecular thickness (Tb.Th, µm), and number (Tb.N, n).

### 2.6. Statistical Analyses

Prism 7 (GraphPad Software Inc., San Diego, CA, USA) and Sigma Plot v13.0 (Systat Software Inc., San Jose, CA, USA) were used for all the statistical analyses and graphs.

## 3. Results

### 3.1. NIPEP-OSS Delays Myeloma Progression in the MM Model

To investigate the effects of NIPEP-OSS on MM growth, we used the improved MM model by injecting 5TGM-1-*Luc* cells in NSG mice as previously described [34]. Through weekly in vivo bioluminescence (BL) imaging, we were able to see that the BL signals above the background, a sign of MM engraftment and growth, started to show from the second week post-injection, as shown in Figure 1B. Although the 5TGM1-*Luc* NSG model has uniform MM progression [34], we randomized the mice based on their BL signals into groups before the drug treatment. Progressive increases in the BL signals were shown, and the focal BL-positive area increased throughout their bodies (Figure 1). Initially, we chose two different concentrations: 24 or 72 mg/kg. The concentration tested in the previous anti-osteoporosis mouse study was 24 mg/kg, and the concentration of 72 mg/kg is three times higher than this since the anti-tumor effect requires a higher dose than that for anti-osteoporosis. However, we did not see any significant difference in MM progression in the mice. Therefore, we chose the 24 mg/kg TIW treatment. In Figure 1, we can see a reduction in the tumor burden in the NIPEP-OSS-treated group from the first week of treatment and throughout the entire test period. In addition, the average of this group of mice was significantly lower than that of the control group. The NIPEP-OSS-treated mice show delayed MM progression compared to that of the control mice. Since the NIPEP-OSS treatment delayed MM progression, we also performed the Mantel–Cox test to see if NIPEP-OSS affects survival. It found that the two groups’ median survival times in this mouse model are significantly different (*p* = 0.0014) from each other, with 45 days in the control group and 57 days in the treatment group. Furthermore, 20% of the NIPEP-OSS-treated mice showed no signs of tumors.

### 3.2. NIPEP-OSS Increases Bone Minerals and Induces Bone Formation in MM Mice

MMBD is one of the major features of MM, which appears in over 90% of MM [28,35,36,37,38]. Myeloma cells disrupt the balance of bone resorption/formation by activating osteolytic devastation, resulting in MMBD [39]. NIPEP-OSS has a selective induction of osteogenic and adipogenic differentiations and shows effective therapeutic activity against osteoporosis [16]. Therefore, the bone minerals in the spine and femurs of MM and control mice were accessed using a PIXImus Densitometer (G.E. Lunar, Madison, WI, USA). Since the mice died on variable days during the trials, a mouse was sacrificed when it met the endpoint criteria, such as hindlimb paralysis, body weight loss (>20%), significant high-BL signals in the whole body, or a body condition score of <2. The mice were sacrificed according to our protocol. The carcasses were stored in the freezer for at least three days. Then, we thawed carcasses, removed the skin, and performed an ex vivo DEXA scan. The DEXA results demonstrated that the bone mineral density (BMD) and bone mineral content (BMC) significantly increased (*p* = 0.01 and *p* = 0.001) on the spine and left/right femurs in the NIPEP-OSS-treated group compared to the control, as shown in Figure 2.

Since this model showed severe bone loss at the spine during MM progression [34], we also performed micro-computed tomography (micro-CT; Scanco Medical AG, Wangen-Brüttisellen, Switzerland) to evaluate the MMBD. For this, the spine was extracted after a DEXA scan, washed with 70% ethanol, stored in PBS for 2–3 days, and then scanned with micro-CT. Our histomorphic analyses on the vertebral bodies found that the NIPEP-OSS treated group had significantly increased trabecular thickness, trabecular number, and bone volume density (*p* < 0.05) compared to the control group (Figure 3A–C).

### 3.3. NIPEP-OSS Showed Similar Effects in C57BL/KaLwRij Model as in NSG MM Model

Radl et. al. first reported a 5T mouse model of myeloma [40]. It has been the most well-characterized syngeneic model and resembles many features of the human benign MGUS stage [41]. Several myeloma cell lines and models have developed from this model [42]. Among these models, the 5TGM1-transplanted C57BL/KaLwRij model has a short latency and develops a tumor burden with pronounced osteolytic lesion formation [43]. Due to such benefits, this model has been extensively used on various drug tests, particularly anti-MMBD [42]. To reconfirm our findings and evaluate an NIPEP-OSS effect on MM growth, we assessed the effects of NIPEP-OSS on MM growth in the 5TGM1-*Luc*-transplanted C57BL/KaLwRij model. Considering the significant interruption in the BL signals by the black fur of these mice, we still performed weekly in vivo BL imaging and show representative mice of each group in Figure 4A. The images in this Figure are set at the same scale as that in Figure 1A to compare these two models, although the presented scale in this figure does not show clear signals in the early stage of myeloma. We also measured the total flux value from the whole body in the C57BL/KaLwRij MM mice model and plotted it in Figure 4B. In both groups, the tumor burdens were detected from the second week post-injection, and the BL signals from the NIPEP-OSS-treated mice started to be differentiated from the control at the fourth week. Thereafter, while the tumor burdens of the control mice continuously increased, the NIPEP-OSS-treated mice gradually decreased or slowed MM progression (Figure 4B). The survival of the treated group was significantly increased (*p* = 0.0057), with a median survival time of 63 days compared to 46 days in the control group (Figure 4C).

We previously found that although both the C57/KaLwRij and NSG MM models developed MMBD, the C57/KaLwRij MM model developed less severe osteolytic lesions than the NSG MM model. To see if NIPEP-OSS also induces bone formation in the C57/KaLwRij MM model, we instead checked a serum bone formation marker, p1NP, to see the whole body rather than locally. Our p1NP analysis shows that serum p1NP levels decreased as myeloma cells were engrafted and proliferated. While the control mice continuously decreased their p1NP levels, the NIPEP-OSS treated mice increased their serum p1NP levels after the treatment (Figure 4D).

## 4. Discussion

MM is the second most common hematologic malignancy, and MMBD is a hallmark characteristic of MM [44,45]. MMBD has devastating consequences for patients, including dramatic bone loss, severe bone pain, and pathological fractures that markedly decrease MM patients’ quality of life and impact their survival. It results from an imbalance of bone formation and resorption. MMBD treatments aim at maintaining patients’ current status rather than healing, by for example using bisphosphonates and denosumab, a humanized monoclonal antibody directed against RANKL [46]. Several studies have recently emphasized the importance of bone regenerative therapy that can reduce the risk of bone-related disease by increasing bone formation, with greater safety, less toxicity, and greater therapeutic effects than anti-resorptive therapy [47]. These drugs showed delayed bone resorption and/or promoted bone formation but had limited effects on MM progression [37].

Recent developments in biomaterials are interesting, particularly in the field of bone regeneration. The CBM peptide has been identified in OPN [48]. This CBM peptide directly binds to collagen due to its binding specificity for the collagen surface. This binding causes active biomineralization and enhances bone remodeling dynamics [5]. Furthermore, the CBM peptide selectively induced osteogenic differentiation, while reducing adipogenic differentiation in human mesenchymal stem cells and inducing bone formation in ovariectomized mice [16].

In this study, we tried repurposing the potent bone-regenerative CBM peptide NIPEP-OSS for multiple myeloma. We hypothesized that NIPEP-OSS may heal MM-induced bone lesions, but also significantly impact MM progression. To test its potential for MMBD therapy, we used a newly improved 5TGM1-engrafted NSG mouse model which had uniform MM development with severe MMBD [34]. Our results demonstrated that NIPEP-OSS not only delays myeloma progression in mice in this model but also irradicates myeloma cells in 20% of mice. We also evaluated bone formation in these NIPEP-OSS-treated mice. The DEXA and microCT results showed significant increases in the BMD, BMC, BV/TV, Tb.N, and Tb.Th in the vertebral body of the NIPEP-OSS-treated skeletons compared to those of the control.

Although the 5TGM1-engrafted NSG mouse model has the benefits described above [34], this model is immuno-deficient, and we know that the immune system functions in various fields of MMBD pathology, including osteoimmunology and crosstalks between immune cells and bone cells [49,50]. Therefore, we also evaluated NIPEP-OSS’s effect in an immune-competent MMBD model, 5TGM1-engrafted C57BL/KaLwRij model. Based on our previous experience, this model has a wide range of MM progression with less severe MMBD than the NSG model. To our surprise, NIPEP-OSS showed a similar improvement in the survival rate as the NSG model, and 20% of the mice were disease-free during the tested period. Instead of DEXA or microCT analyses, we measured the serum bone formation marker p1NP in these animals and found that the p1NP levels were increased immediately after the NIPEP-OSS treatment, while the control mice continuously decreased their serum level as MM progressed. Therefore, NIPEP-OSS enhanced bone formation by increasing biomineralization in bone.

Our results here provide the first insight into the use of collagen–CBM complex in MM therapy by restoring bone reabsorption and improving the overall survival of MM mice in NIPEP-OSS-treated groups. Our results show that a potent bone anabolic agent heals bone lesions but also stalls MM progression.

## 5. Conclusions

The current study used a potent bone anabolic agent, NIPEP-OSS, for MMBD therapy. In two preclinical models, our results showed that NIPEP-OSS improves bone formation and suppresses MM progression. Although further investigations are necessary as it is a novel clinical therapy, our success in repurposing the CBM peptide for MM therapy may demonstrate its potential as a new therapeutic drug for MMBD.

## Figures and Tables

**Figure 1 cancers-15-02473-f001:**
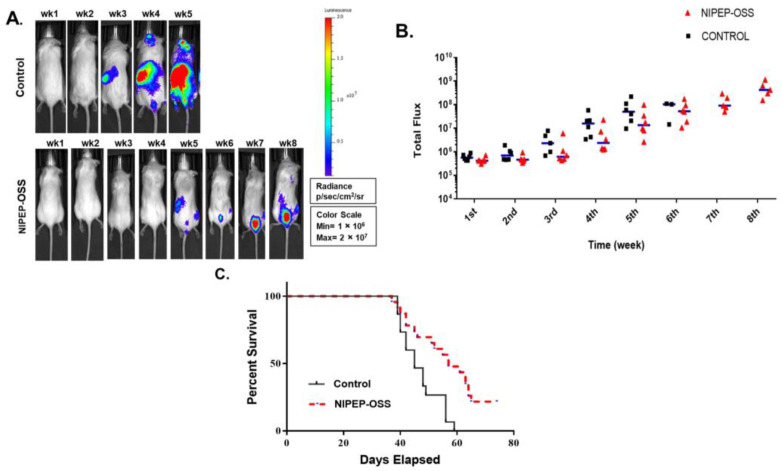
NIPEP-OSS delays myeloma progression. (**A**) Representative BL images of PBS vs. NIPEP-OSS-treated 5TGM-1 transplanted NSG mice. All BL images were normalized in a single scale, as shown in the box. (**B**) Total flux values were obtained from the whole body in the weekly image. All total flux values (black square or red triangle; PBS or OSS) and medians (blue line) indicate myeloma development. (**C**) Mouse survivors were censored daily. When a mouse met endpoint criteria, it was sacrificed and recorded. Kaplan–Meier plot was plotted from the records of 15 mice from PBS (black line) and 23 mice from NIPEP-OSS (red dotted line). Median survival times were 45 days for PBS and 57 days for NIPEP-OSS-treated mice. The Mantel–Cox test gave a significance of *p* = 0.0014.

**Figure 2 cancers-15-02473-f002:**
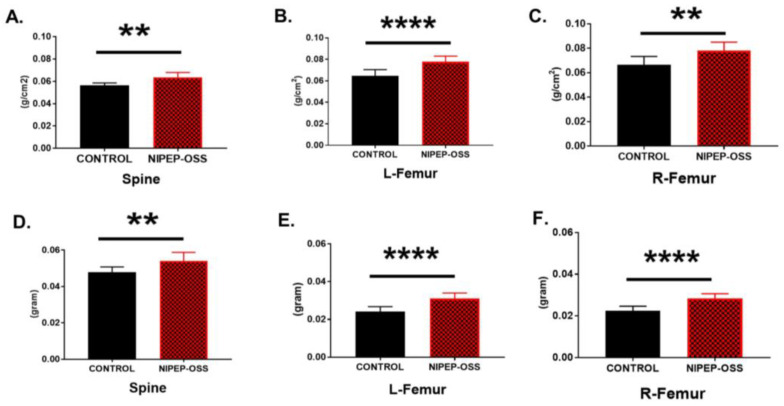
NIPEP-OSS increases both bone mineral density and content. After meeting the endpoint criteria, mice were sacrificed and stored at −20 °C (>2 days). Then the carcasses were thawed, the skin was removed, and a DEXA scan was performed. (**A**–**C**) show the bone mineral density (BMD) changes in the spine, left femur, and right femur of NIPEP-OSS mice versus control (PBS-treated). (**D**–**F**) show the bone mineral content (BMC) changes in NIPEP-OSS mice versus control mice in the spine, left femur, and right femur. BMD and BMC values with standard error are expressed here from the control versus NIPEP-OSS groups at each tissue, and statistical differences were analyzed using Student’s *t*-tests. ** denotes *p* < 0.01.**** denotes *p* < 0.0001.

**Figure 3 cancers-15-02473-f003:**
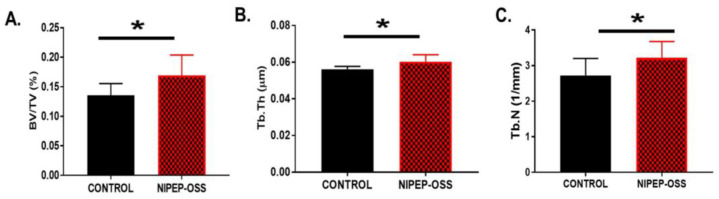
NIPEP-OSS increases BV/TV, Tb.N, and Tb.Th. After DEXA scanning, the spine was extracted and washed/stored in PBS. The spine was scanned by micro-CT. On the bone histomorphic analyses, we analyzed the trabecular regions of total lumbar vertebrae (L1–L6) and expressed (**A**) bone volume (BV/TV), (**B**) trabecular thickness (Tb.Th), and (**C**) trabecular number (Tb.N) in each group as control vs. NIPEP-OSS. Means with with standard error are expressed here from the control versus NIPEP-OSS groups and statistical differences were tested using Student’s *t*-tests. * denotes *p* < 0.05.

**Figure 4 cancers-15-02473-f004:**
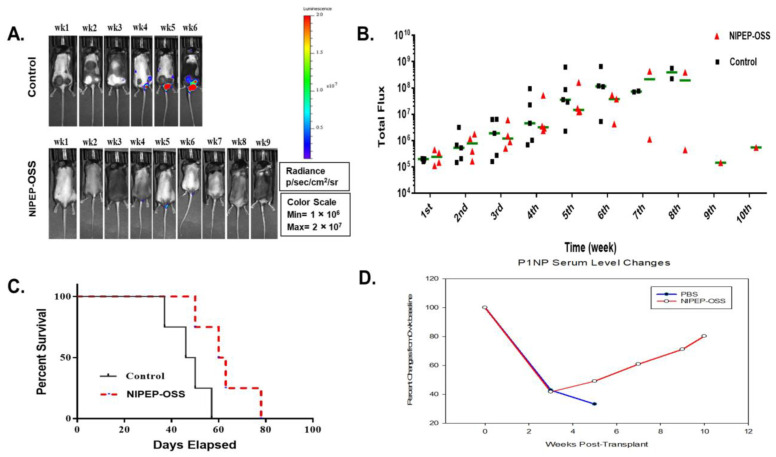
NIPEP-OSS also delays myeloma progression in the C57BL/KaLwRij model. (**A**) BL images of 5TGM-1-transplanted C57BL/KaLwRij mice. All BL signals were normalized on a single scale. (**B**) The weekly images obtained total flux values from the whole mouse body. All total flux values (black square or red triangle) and medians (green line) were expressed to show progressions of myeloma in both groups. (**C**) Surviving mice were censored daily. When a mouse showed endpoint criteria, the mouse was sacrificed and scored. Median survival was 46 days for PBS compared to 63 days for NIPEP-OSS-treated mice. The Mantel–Cox test showed a significance of *p* = 0.0057. (**D**) p1NP levels were obtained from plasma using a p1NP ELISA kit. Means of serum p1NP levels were analyzed and expressed from each PBS (red line with open circle) and NIPEP-OSS (blue line with closed circle) group.

## Data Availability

The data stated in this study are available upon request from the corresponding author.

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
