# Peer review of "The Synthetic Collagen-Binding Peptide NIPEP-OSS Delays Mouse Myeloma Progression"

_cancers, 2023, doi:10.3390/cancers15092473_

Round 1

Reviewer 1 Report

Authors propose the efficacy of a bone anabolic agent in the treatment of Multiple Myeloma. The results look promising. A clearer or better resolution version of figure 4 will be appreciated.

As far as the scientific soundness of the manuscript goes, it is well planned.  Of course, the effects are mild but then again, the overall manuscript goes well with the special edition on 'repurposing the drugs for cancer treatment'.  The authors use a bone anabolic agent to show that it can delay progression of multiple myeloma in a mouse model.  Their data suggest that diseases such as multiple myeloma are niche dependent, and niche augmentation can help the current treatment in combinatorial strategy.

One can always ask for more experiments with combinations of current therapeutic agents and this peptide, but that could be out of the scope of the current manuscript. 

Author Response

We appreciate your great comments and gave us positive feedback.

Investigating the combinatory effect is well-accepted. We will perform it at the next set of experiments.

Reviewer 2 Report

This is a very well presented manuscript. The study design appears to be sound, albeit with some surprise that the authors choose not to compare NIPEP-OSS with existing therapies, in particular biphosphonates such as clodronate, pamidronate and zoledronic acid. Have they any evidence that it is superior to the latter group of drugs?

Further questions that need to be addressed in future mouse model experiments and, of course, studies in humans include potential toxicity of NIPEP-OSS. In the first instance, however, this is an interesting and well presented study. 

Author Response

Thank you very much for the positive feedback. 

The toxicity study of the NIPEP-OSS was conducted previously for their commercialization. The following were the results we obtained from NIBEC;

The safety study was conducted at GLP (Good Laboratory Practice) agency using mouse and monkey. No systemic toxicity was observed at 200mg/kg/day NIPEP-OSS in mouse and monkey when it was injected once daily for 4 weeks. The human equivalent dose was calculated by FDA guideline and 1620mg [200X0.135 (mouse conversion factor)=27mg/kg/day] and 3840mg [200X0.32 (monkey conversion factor)=62mg/kg/day] for 60kg human. Taken together, NIPEP-OSS has enough safety for human study.

Reviewer 3 Report

The manuscript “The Synthetic Collagen Binding Peptide, NIPEP-OSS, delays 1 Mouse Myeloma Progression” is well organized; the presentation of methods, results and discussion are adequates.

The paper results well written and it is of general interest for both clinical and research readership, therefore, the overall evaluation for the manuscript is satisfying

Author Response

We appreciate great and positive feedback regarding our manuscript.